# Optimization of the Brewing Conditions of Shanlan Rice Wine and Sterilization by Thermal and Intense Pulse Light

**DOI:** 10.3390/molecules28073183

**Published:** 2023-04-03

**Authors:** Xiaoqian Wu, Yunzhu Zhang, Qiuping Zhong

**Affiliations:** 1School of Food Science and Engineering, Hainan University, Haikou 570228, China; 2Key Laboratory of Food Nutrition and Functional Food of Hainan Province, Haikou 570228, China; 3School of Biomedical Engineering, Hainan University, Haikou 570228, China

**Keywords:** Shanlan rice wine, free amino acids, volatile compound, pasteurization, intense pulse light sterilization

## Abstract

This study aimed to optimize the brewing conditions of Shanlan rice wine (SRW) and select a suitable sterilization method. The response surface method experiment was used to optimize the brewing process of SRW. LC-MS/MS (liquid chromatography–tandem mass spectrometry) and GC-MS (gas chromatography–mass spectrometry) were used to analyze the physicochemical components, free amino acids, and flavor metabolites of the thermal-sterilized SRW and the SRW sterilized by intense pulsed light (IPL), respectively. Results showed that the optimum fermentation conditions of SRW were as follows: fermentation temperature, 24.5 °C; Qiuqu amount (the traditional yeast used to produce SRW), 0.78%; water content, 119%. Compared with the physicochemical properties of the control, those of the SRWs separately treated with two sterilization methods were slightly affected. The 60 s pulse treatment reduced the content of bitter amino acids, maintained sweet amino acids and umami amino acids in SRW, and balanced the taste of SRW. After pasteurization, the ester content in wine decreased by 90%, and the alcohol content decreased to different degrees. IPL sterilization slightly affected the ester content and increased the alcohol content. Further analysis of the main flavor metabolites showed that 60 s pulse enhanced the important flavor-producing substances of SRW. In conclusion, 60 s pulse is suitable for sterilizing this wine.

## 1. Introduction

The Li people in Hainan Province, China, are an indigenous minority with a 3000-year history. In the long development process, the Li ethnic people have created a splendid and colorful national culture. Shanlan rice wine (SRW) brewing presents many regional characteristics from their culture. SRW, also known as “biang” wine in the local language, enjoys a high reputation in Hainan and is highly respected by the locals.

SRW is made by the Li nationality people using a traditional starter culture called “Qiubing” or “Qiuqu.” It has a unique aroma, subtle flavor, low alcoholicity, sweet taste, mild acidity, and good vinosity. An SRW aged 3 to 5 years possesses elegant flavor and mellow taste [1]. In the traditional brewing process of SRW, the Qiubing used is made in an open, nonsterile environment. It also contains various microorganisms (yeast, mold, and bacteria). Thus, its use often leads to unstable product quality.

In the brewing process of SRW, thermal sterilization is the most critical step in ensuring food safety and extending the product’s shelf life. Many sterilization methods exist, and each has advantages and disadvantages. Currently, nonthermal sterilization technologies, such as intense pulsed light (IPL) technology [2], high hydrostatic pressure technology [1], ultrasound technology [3], pulsed electric field technology [4], irradiation technology [5], and flash pasteurization [6] have attracted much attention. The IPL is a fast and residue-free technology for removing surface contamination of food without damage caused by heat [7,8]. Bhagat and Chakraborty (2022) compared untreated, pasteurized, and IPL-treated pomegranate juice. They found that IPL sterilization can ensure the safety of microorganisms and effectively preserve the nutritional value of pomegranate juice. The destructive impact of IPL sterilization on the microbial and enzyme activities in juice is less than that of a thermal treatment while successfully preserving the antioxidant properties of mixed fruit beverages [9].

However, the application of IPL to Chinese rice wine has received little attention. Moreover, the preparation of pure-cultured Qiuqu for brewing SRW and the application of IPL to SRW have not been reported. Therefore, the present study aimed to optimize the brewing conditions of SRW using the response surface method (RSM). This study also explains the changes in the physicochemical properties, free amino acids, and volatile components of the SRW treated with IPL compared with those of the SRW treated with thermal sterilization.

## 2. Results

### 2.1. Optimization of Fermentation Conditions of SRW

#### 2.1.1. Single-Factor Experiment Results

Figure 1a shows the sensory score and the alcohol, total sugar, and total acid contents of the brewed SRW at different fermentation temperatures with a fixed Qiuqu addition amount of 1.0% and water content of 120%. The alcohol content of SRW increased slowly with the increase in temperature before 26 °C. It reached its highest at 26 °C and then remained stable. The fermentation temperature lower than 26 °C resulted in the poor taste of SRW, with low total sugar content and high total acid content caused by low yeast activity and incomplete fermentation. The sensory score reached the highest when the fermentation temperature was 26 °C. Thus, 26 °C was regarded as the optimum fermentation temperature.

Figure 1b shows the sensory score and the alcohol, total sugar, and total acid contents of the brewed SRW with different water contents with a fixed fermentation temperature of 26 °C and Qiuqu amount of 1%. The alcohol content of SRW first increased rapidly with the increase in water content and then decreased rapidly. In the fermentation process, the low water and high sugar contents inhibited the growth and propagation of yeast. Thus, sugar could not be fully converted into alcohol, resulting in the poor taste of SRW with an excessively high total sugar content. When the water content was 150%, the fermented SRW with the highest sensory score had a good gloss, soft taste, and typical flavor. Thus, the water content of 150% was regarded as the optimum treatment condition. Figure 1c shows the sensory score and the alcohol, total sugar, and total acid contents of the brewed SRW at different Qiuqu addition amounts with a fixed fermentation temperature of 26 ℃ and water content of 120%. The alcohol content of SRW increased with the increase in the amount of Qiuqu. When the amount of Qiuqu was 1%, the ratio of sugar and acid in the wine body was moderate, the sour and sweet were in harmony, and the sensory score was the highest. Thus, 1% of Qiuqu was considered the optimum amount for SRW brewing.

#### 2.1.2. Response Surface Model Analysis of Sensory Evaluation

In the present study, response surface optimization was used to determine the optimal fermentation conditions of SRW. The sensory evaluation of fermented products was conducted to evaluate the effect of different fermentation conditions on the sensory performance of SRW. The final regression equation of the response was evaluated according to the experimental design software. The second-order quadratic equation was used to represent the function quadratic polynomial model of Y (sensory evaluation) as the coding independent variable as follows:Y = 79.46 + 2.49X_1_ − 0.97X_2_ + 0.71X_3_ + 1.52X_1_X_2_ + 0.3X_1_X_3_ − 0.18X_2_X_3_ − 4.86X_1_^2^ − 2.83X_2_^2^ − 3.55X_3_^2^

In the formula, X_1_, X_2_, and X_3_ are the independent variables of fermentation temperature (°C), Qiuqu amount (%), and water content (%); Y is the response value.

According to the design of the experiment, Table 1 shows the main influence factors (sensory evaluation), the values of total acid, total sugar, and alcohol precision of the 17 groups of experiments in response to surface optimization, and the analysis of variance of the response surface secondary model. The fitness of the model was studied through the *p* value, correlation coefficient (R^2^), and the lack of fit of the model. As shown in Table 1, the variance analysis model of response results (*p* < 0.0001) proves that the model is extremely significant. Moreover, R^2^ = 98.99, and the difference between R^2^adj and R^2^pre is less than 0.2. This finding demonstrates that the model is well-fitted, and the lack of fit is not significant, indicating that the regression equation prediction obtained from the test can reasonably predict the response value. Therefore, it can be used to analyze the test results.

#### 2.1.3. Response Surface Optimization of the Interaction between Various Factors

Figure 2 shows the response surface diagram of the influence of three factors (fermentation temperature, Qiuqu amount, and water content) on the sensory score of SRW to analyze the interaction between various factors thoroughly. Figure 2a shows the three-dimensional response surface diagram of the sensory score changing with fermentation temperature and the amount of Qiuqu. The sensory score initially increased with the increase in the amount of Qiuqu. It began to show a slow decline when it reached the maximum value. The sensory score increased when the fermentation temperature increased. However, the sensory score decreased when the fermentation temperature exceeded 24 °C. This finding indicates that the high temperature inhibited the growth and propagation of microorganisms, thereby affecting the flavor of SRW [10]. Figure 2b clearly shows the interaction of fermentation temperature and water content on sensory evaluation. According to the three-dimensional response surface graph, the sensory score increased with the increase in fermentation temperature and water content and decreased with the increase in fermentation temperature and water content after reaching the maximum value. This result is consistent with previous research results [11]. Finally, Figure 2c analyzes the interaction of water content and the amount of Qiuqu on sensory scores. The score increased first under the same fermentation temperature. Then, it decreased with the water content and the amount of Qiuqu. This phenomenon occurred possibly because the utilization rate of raw materials increased with the increase in Qiuqu content. Moreover, other flavor substances were produced. However, the yeast autolysis phenomenon occurred with the extension of fermentation time, leading to the premature end of fermentation, which affected SRW quality. Moreover, the water content was too little to maintain the normal physiological activities of microorganisms [12].

The optimal fermentation conditions of SRW with the highest sensory score are as follows: fermentation temperature, 24.48 °C; Qiuqu amount, 0.78%; water content, 118.96%; sensory score predicted value under the optimized conditions, 79.9. The formula was adjusted as follows to test the reliability of the optimized conditions: fermentation temperature, 24.5 °C; Qiuqu amount, 0.78%; water content, 119%. The adjusted formula was used to perform validation tests. Three parallel groups were used for each test. The sensory scores of the tests were 81, 79, and 78.8, with an average of 79.6. The experimental results fit well with the model.

### 2.2. Effect of Sterilization Treatment on the Physicochemical Properties of SRW

The physicochemical properties (total acid, total sugar, pH, and alcohol) of control and pasteurized and IPL-treated SRW are shown in Table 2. After sterilization, the pH value of SRW increased, but the total acid decreased. This finding is consistent with the study by Jin, indicating that different sterilization method treatments can decrease the total acid and increase the pH of rice wine. This phenomenon occurred possibly because lactic acid bacteria lead to lactic acid production in rice wines without a sterilization treatment. After sterilization treatment, the number of lactic acid bacteria in the rice wine samples was greatly reduced, resulting in reduced total acid content in the wine [13]. The changes in the total acid and pH value of the SRW treated with IPL were more considerable than those of the pasteurization samples. This phenomenon occurred possibly because IPL is better than heating sterilization in destroying the activity of some microorganisms in yellow rice wine [14]. The amylases in the starter hydrolyze starch in the raw material to produce reducing sugar; moreover, yeast reproduces and grows by using fermentable sugar to produce alcohol [11]. Table 1 shows that compared with the control, the IPL-treated SRW exhibited decreased total sugar content (*p* > 0.05) and alcohol content (*p* < 0.05), whereas the thermal processing-treated SRW showed increased total sugar content (*p* < 0.05) and decreased alcohol content (*p* > 0.05). These results indicate that the effects of the thermal treatment on the inactivated residual microorganisms were more than those of the IPL treatment. IPL treatment leads to ethanol volatilization, which may be related to the open environment during treatment.

### 2.3. Effects of Different Sterilization Treatments on the Free Amino Acids in SRW

The amino acid in SRW is mainly produced via protein hydrolysis in raw materials by the protease in microorganisms in the starter culture [15,16]. The amino acid in SRW is much more than that in other alcoholic beverages A total of 21 free amino acids identified in SRW are shown in Table 3. Asn, Gln, Gly, Glu, Ala, Pro, and Leu are the main components of all amino acids in SRW. In the three groups of pasteurized SRW, the content of the total free amino acids in Rw-past65 increased. However, the opposite change was demonstrated in the two other groups. The total free amino acids in Rw-past75 were the lowest among the three groups of pasteurization, possibly because of the excessive loss of amino acids in rice wine caused by the Maillard reaction at high temperatures and long periods [1]. The loss of amino acid content in Rw-past85, which was less than that of Rw-past75, was relative to the weakened Maillard reaction when the reaction system temperature was higher than 80 °C. However, the longer the reaction time is, the greater the influence of the Maillard reaction is. Similar decreasing trends of amino acids were also observed in the IPL-treated SRW.

Amino acids can be classified into bitter, sweet, and umami amino acids according to their taste characteristics. Bitter amino acids include Arg, His, Lys, and Try; sweet amino acids include Ser, Gly, and Met; umami amino acids include Asp and Glu [17,18]. The changes in the total contents of flavorful amino acids and free amino acids in SRW before and after sterilization are shown in Table 3. In general, the sweet amino acids dominated the SRW before and after sterilization treatment. Moreover, the thermal treatment of 65 °C increased the sweet and bitter amino acid contents of SRW. However, a significant decrease in these contents was observed in Rw-past75, Rw-Past85, Rw-Pulse C, Rw-past75, Rw-pulse B, and Rw-pulse C. Compared with the control, the SRW treated with 65 °C showed a decreased total amino acid content. This phenomenon may be caused by the decomposition of proteins in SRW caused by long-term sterilization. Some small molecules of flavoring peptides were also produced. In general, a 60 s pulse can reduce the bitter amino acid content in SRW and maintain the sweet and umami amino acid contents in SRW. These results indicated that the 60 s pulse had a good effect on SRW quality.

The differences in 21 free amino acids between the unsterilized and other sterilized SRW are shown in Figure 3. The results showed that the free amino acids in the SRW after sterilization had remarkable changes. The decrease in the content of free amino acids in cold sterilization treatment was less than that in heat sterilization treatment. Pasteurization increased the temperature and resulted in excessive loss of nutrients in the wine. Cold sterilization did not need to heat the rice wine and used physical conditions to kill the microorganisms in the wine [19], thereby reducing the damage to the nutritional components in the rice wine The contents of Leu, Val, Pro, Ala, Glu, GABA, Lys, Asp, Gly, and Thr in the treated SRW were higher than those in the control. This finding indicates that the sterilization conditions at low temperatures are mild, and the amino acids in SRW can be retained well. Figure 3 shows that the amino acid content in Rw-past65 and Rw-pulse A had a small range of change, whereas the other sterilization groups had a considerable impact on the amino acid content, particularly in Rw-past75 and Rw-past 85. This phenomenon shows that a high sterilization temperature results in a significant reduction of amino acid content in SRW. In Rw-pulse A, Arg, Thr, and 4-hydroxy-L-proline were more than 0.8 μg/mL; in Rw-past65, except for Phe, the contents were more than 0.8 μg/mL; in Rw-past85, only the Phe content was more than 0.8 μg/mL. The amino acid content of the remaining three sterilization groups (Rw-past75, Rw-pulse B, and Rw-pulse C) did not exceed 0.8 μg/mL. The amino acid content decreased dramatically in Rw-pulse C, indicating that the sterilization time was too long. The amino acid content in Rw-past65 increased, whereas that in Rw-past75 and Rw-past85 decreased. This finding indicates that the selection of temperature greatly influences the amino acids during pasteurization. The contents of 4-hydroxy-L-proline, try, and tyr showed the greatest changes in Rw-pulse B. In comparison, leucine, ile, and gly showed the greatest changes in Rw-pulse C and Rw-past85. Moreover, his and glu were sensitive to 90 s pulse and 85 °C treatment, respectively.

### 2.4. Difference of Flavor Metabolites before and after Sterilization by PCA

#### 2.4.1. PCA

PCA is a statistical analysis method of multidimensional data with unsupervised pattern recognition. This analysis method is often used for studying how the internal structure among multiple variables is revealed through a few principal components. In the present study, the PCA method was used for analyzing the main volatile compounds in different SRWs to understand the total metabolite difference between the control group and the experimental group and the variation degree between the samples in the group. PCA was also used for analyzing the difference in metabolites between the experimental group (after sterilization) and the control group (without sterilization). Figure 4a shows that the contribution rates of the first principal component (PC1) and the second principal component (PC2) between the Rw-past65 and control groups were 38.03% and 26.05%, and the total contribution rate was 64.08%. The PC1 showed that both groups were close to the center line, whereas the PC2 demonstrated that both groups were located on the positive axis. These findings indicate a difference between the two groups, mainly in PC1. Figure 4b,c show that the contribution rates of the PC1 and the PC2 between Rw-past75 and control were 57.98% and 21.23%, respectively, and the total contribution rate was 79.21%. The contribution rates of the PC1 and the PC2 between the Rw-past85 and control groups were 40.17% and 30.82%, respectively, and the total contribution rate was 70.99%. Rw-past75 and Rw-past85 were located on the negative axis of PC1, whereas the control was located on the positive axis of PC1. From the direction of PC2, Rw-past75, Rw-past85, and the control group were near the center line. The PC1 of t Rw-past75, Rw-past85, and the control group showed intergroup differences, whereas PC2 showed intragroup aggregation. This finding indicates that the metabolic differences between the two groups of treatment and control groups were mainly reflected in PC1. All samples were within a 95% confidence interval, and the degree of separation between groups was good. This finding indicates differences in the metabolism between the control group and two groups of treatment, namely, Rw-past75 and Rw-past85. Figure 4d–f show that the IPL treatment group was more similar to the control group than the pasteurization group was. Among the three pulse sterilization duration groups, the 30 s and 90 s groups showed sterilization duration highly similar to that of the control group.

#### 2.4.2. Screening and Analysis of Metabolites of Flavor Difference in SRW before and after Sterilization Treatment

Flavor substances affect the taste and flavor of yellow rice wine [20]. The volatile substances in wines mainly come from raw materials or are produced by microorganisms and enzymes in the koji. However, pasteurization and pulsed light sterilization treatment still impact flavor substances in yellow rice wine. In the present study, the SPME-GC-MS method was used to conduct qualitative and quantitative analysis of metabolites in SRW before and after sterilization. A total of 407 metabolites were detected in seven groups of wine samples. The composition of metabolites is shown in Figure 5. The distribution of the major metabolites in the samples can be investigated as a whole by analyzing the metabolite composition ratio. As secondary metabolites with a low content in yellow rice wine, volatile aroma compounds, such as higher alcohols, esters, phenols, aldehydes, and ketones, considerably contribute to flavor; however, their odor activity threshold is low [21,22]. Figure 5 shows that among the metabolites (alcohols, esters, acids, aldehydes, ketones, and phenols) with a great influence on the flavor of SRW, ester compounds accounted for the largest proportion (16.46%), followed by alcohol compounds (9.34%). Phenolic substances (1.97%) had the least proportion. Acids, ketones, and aldehydes were 3.19%, 6.63%, and 7.37%, respectively. Based on OPLS-DA analysis, differential metabolites were further screened by combining the VIP value of ≥1 and differential change value of ≥2 or ≤0.5. The volcano plot of the screened differential metabolites is shown in Figure 6. Each dot in the figure represents a metabolite, with the green dots representing downregulated differential metabolites, the red dots representing upregulated differential metabolites, and the gray dots representing metabolites that are detected but not remarkably different. The ordinate represents the VIP value. The larger the ordinate value is, the more significant the difference is, and the more reliable the differential metabolites obtained by screening are. Figure 6a shows that the control and Rw-past65 groups had 32 differential metabolites, of which 9 were upregulated, representing an increase in relative content, whereas 23 were downregulated. This finding indicates a decrease in relative content. Further analysis showed that the differential metabolites of the control and Rw-past65 groups accounted for 7.9% of the total differential metabolites. Figure 6b shows that the control and Rw-past75 groups had 33 differential metabolites, of which 11 were upregulated, whereas 22 were downregulated. Moreover, the differential metabolites accounted for 8.1% of the total different metabolites. Figure 6c shows that the control and Rw-past85 groups had 35 differential metabolites, of which 12 were upregulated, whereas 23 were downregulated. Moreover, the differential metabolites accounted for 8.6% of the total differential metabolites. Figure 6d shows that the control and Rw-pulse A groups had three differential metabolites, among which three were upregulated, and none were downregulated. Moreover, the differential metabolites accounted for 0.74% of the total differential metabolites. Figure 6e shows that the control and Rw-pulse B groups had 15 differential metabolites, among which 14 were upregulated, and 1 was downregulated, accounting for 3.7% of the total different metabolites. Figure 6f shows that the control and Rw-pulse C groups had four differential metabolites, among which four were upregulated, and none were downregulated. Moreover, the differential metabolites accounted for 0.98% of the total differential metabolites. The above results showed that compared with the rice wine samples treated with pulse sterilization, those treated with pasteurization had noticeably different metabolites. However, compared with the two other groups, the 60 s pulse group produced upregulated differential metabolites. This finding indicates that the 60 s pulse treatment may increase the flavor of SRW.

On the basis of database comparison, the metabolites of SRW before and after sterilization were identified by referring to the relevant literature to understand further the influence of sterilization on specific flavor components of SRW. These metabolites include esters, alcohols, aldehydes, ketones, and phenols, as shown in Table 4. As one of the most important flavor components in SRW, ester compounds are mainly obtained by esterifying organic acids and amino acids with alcohols under the action of acetyltransferase, which can provide a satisfactory fruity flavor of SRW, given its low threshold concentration and ideal fruit flavor, the quality of fermented beverages is affected [23,24,25]. Compared with the control group, the three pasteurized groups (Rw-past65, Rw-past75, and Rw-past85) exhibited a decreased total relative peak area of ester compounds by approximately 90%. The total relative peak area of the three IPL sterilization groups (Rw-pulse A, Rw-pulse B, and Rw-pulse C) almost did not change. This finding indicates that pasteurization can greatly reduce the ester content in wines. The reason for the above phenomenon may be that after pasteurization, some ester substances were hydrolyzed, leading to a decrease in the relative content of ester compounds. IPL sterilization does not produce a high temperature to promote the action of ester substances; thus, it does not have a great effect on the content of ester substances in wines. Ethyl octanoate produced an alcoholic odor, and its content was significantly reduced under pasteurization. However, its content did not have a significant change after IPL sterilization. SRW contains a large amount of alcohol and is the main by-product in the fermentation process of SRW. As the flavoring substance in SRW, alcohols have a rich wine sense and the function of helping aroma; it is also the precursor of an ester substance [26]. The main differential alcohol metabolites identified from SRW are 2-ethyl-1-dodecanol, 2,6-dimethyl-1-nonen-3-yn-5-ol, and (1α, 2α, 3α); 2-methyl-3-(1-methylethenyl) -cyclohexanol, one of the main alcohols in SRW, can provide unique sweetness and floral flavor to SRW [27]. Table 4 shows that 1,3-dioxolane-2,2-diethanol was not detected in the pasteurized SRW. However, its relative content treated with IPL sterilization remained almost unchanged compared with the relative content of the Rw-control group. Moreover, the relative content of alcohols in SRW after IPL sterilization increased. In particular, the relative content of Rw-pulse B increased significantly. After pasteurization, the relative content of alcohol in the wine samples decreased by approximately 20%. This result may be caused by the fact that alcohols form aldehydes and ketones after oxidation reactions. The odor activity value of these substances was relatively low. However, these substances can have certain positive effects on the aroma of SRW. Moreover, they are easy to oxidize into carboxylic acids after sterilization treatment because of their unstable chemical properties, resulting in the reduction of their contents [28]. The alcohol content in Rw-pulse B was remarkable higher than that in the two other groups. The carbonyl compounds detected in SRW were ketones and aldehydes. The content of ketones decreased by approximately 60% after pasteurization but slightly changed after pulse sterilization. The content of ketones increased with the increase in sterilization time. The reason for this phenomenon was that the content of the largest ketones, 1-hepten-3-one, decreased substantially after pasteurization treatment. However, only a small amount of loss was generated after the IPL treatment. The relative content of aldehydes showed an increasing trend after IPL sterilization and pasteurization treatment. The aromatic compound benzaldehyde is an important aroma component in SRW. The relative content of benzaldehyde increased significantly after IPL sterilization, particularly that of Rw-pulse B, which increased by 131%. The possible reason is that in the process of IPL sterilization, the enzyme reaction made benzaldehyde continue to release, whereas the high-temperature pasteurization made the enzyme inactivated and unable to continue to react. The flavor of SRW is mainly produced by esters and alcohols. Thus, pasteurization can reduce the content of esters and alcohols in wines, whereas IPL sterilization can preserve and increase the content of flavor substances in wines. This finding shows that IPL sterilization is superior to pasteurization in preserving the volatile aroma components of SRW. However, in the IPL sterilization group, 60 s pulse indicated an increase in the important aroma components in SRW. Thus, a 60 s pulse had the best effect. In particular, it could enhance the flavor components of SRW and retain the aroma well.

## 3. Materials and Methods

### 3.1. Chemicals and Reagents

Shanlan rice was produced in Wuzhishan City, Hainan Province, China. *Saccharomyces cerevisiae* (GDMCC 2.128) and *Rhizopus oryzae* (CGMCC 3.866) were purchased from Guangdong Microbiological Culture Preservation Center, Guangzhou city, Guangdong Province, China. Methanol, acetonitrile, and formic acid were purchased from ANPEL. Hydrochloric acid was purchased from Sinoreagent. AccQ•Tag reagent was purchased from Waters, Milford, USA. The ultrapure water was prepared in-house using the Milli-Q water purification system (Millipore, Bedford, MA, USA).

### 3.2. Qiuqu Preparation and SRW Brewing

The water content of 100 g bran was adjusted to 73% and sterilized at 121 °C for 40 min. After being cooled, *S. cerevisiae* and *R. oryzae* were inoculated and cultured at 28 °C for 3.5 days. The mature Qiuqu was used as the starter culture of SRW. A certain amount of Shanlan rice was weighed and soaked in water for 5–7 h. The water was drained, and the rice was cooked for 1 h. The cooked rice was placed on a sieve net, cooled to approximately 25 °C, and weighed. A certain amount of Qiuqu was added to the cooked rice. The cooked rice with Qiuqu was stirred well, covered with gauze, and saccharified in an incubator at 28 °C for 2 days. After saccharification, pure water was added to continue the alcohol fermentation. The alcohol fermentation was finished 5 days later. The wine sample was obtained by filtration and centrifugation.

### 3.3. Single-Factor and Response Surface Experiments

The single-factor test was used to study the influence of fermentation temperature (22, 24, 26, 28, and 30 °C), Qiuqu amount (0.6%, 0.8%, 1.0%, 1.2%, and 1.4%), and water content (80%, 100%, 120%, 150%, and 180%) on the winemaking characteristics and SRW quality. Based on a single-factor test, the Box–Behnken test with three factors and three levels of each variable was used to treat 17 experimental groups and optimize the brewing process of SRW further. The independent variables of the experimental design were fermentation temperature (22, 24, and 26 °C), Qiuqu amount (0.6%, 0.8%, and 1.0%), and water content (80%, 115%, and 150%), corresponding to coding levels (−1, 0, 1), (−1, 0, 1). The best fermentation conditions were determined based on the sensory score of the fermented SRW. The physicochemical indexes under different fermentation conditions were also determined.

### 3.4. Sensory Evaluation Method of SRW

The sensory evaluation was conducted with the official methods of GB/T 13662-2018 in China with some modifications. Ten food professionals (five men and five women) with food sensory evaluation training and certain discrimination differences were selected to form an evaluation team and conduct a sensory evaluation on all indicators of rice wine products during the experiment. The comprehensive score of all indicators was taken as the total score. The sensory evaluation was conducted in the sensory evaluation room with a room temperature of 25 °C and relative humidity of 60%. The prepared sample was stored at a controlled temperature using a thermostat to ensure the uniformity of the sample temperature. A 50 mL disposable plastic cup was used to hold samples. Each sample volume was 25 mL. Before the sensory evaluation, five SRW samples were divided into groups. The samples in each group were numbered and randomly distributed to the evaluators. At the end of each group of evaluation, the evaluator rinsed his/her mouth and waited for 5 min before the next group of evaluation to avoid residual aftertaste. The team members rated the appearance (0–15 points, where 0 meant light yellow with sediment and lack of luster; 15 meant yellow, clear, and glossy), aroma (0–30 points, where 0 meant poor flavor and 30 meant harmonious and strong), taste (0–40 points, where 0 meant sour and bitter taste and 40 meant mellow, delicate and well-balanced), and style (0–15 points, where 0 meant poor style and 15 meant typical style ) of SRW.

### 3.5. Sterilization of SRW

The wine samples were exposed to intense pulsed light (IPL) using an IPL system (UV200-2; Limeigu (Shenzhen) Photographic Technology Co., Ltd.; China). The dish with 4–5 mm-thickness SRW was placed at the center of an IPL cabinet. The distance of the xenon lamp with a frequency of 5 Hz and a voltage class of 11 was 2.5 cm. Intense pulse light intensity 1.38 (w/cm^2^). The dish was treated for 30 s (Rw-pulse A), 60 s (Rw-pulse B), and 90 s (Rw-pulse C). The pasteurization treatment conditions were set at 85 °C for 10 min (Rw-past 85), 75 °C for 15 min (Rw-past 75), and 65 °C for 20 min (Rw-past 65) in a water bath. The SRW samples without pasteurization and pulse treatment were used as control.

### 3.6. Determination of SRW’s Physicochemical Properties

Total acid (tartaric acid, g/L), total sugar (glucose, g/L), and alcohol content (%, vol) were determined according to the official analytical methods in China (GB/T13662-2018 and GB/T15038-2006). The pH value was determined with the pH meter method.

### 3.7. Determination of Free Amino Acids in SRW

#### 3.7.1. Sample Preparation and Extraction

The samples were extracted with 0.6 mL 0.1 M hydrochloric acid for 1 h with gentle agitation on a shaker at room temperature. Each sample was filtered through a 0.22 μm pore membrane filter. Then, 10 μL of the sample was taken into a UHPLC vial and added with 70 μL borate buffer and 20 μL AccQ•Tag reagent. The reaction mixture was kept at room temperature for 1 min and heated at 55 °C for 10 min. After cooling, 1 μL was injected.

#### 3.7.2. UPLC Conditions

The sample extracts were analyzed using a UPLC–Orbitrap-MS system (UPLC, Vanquish; MS, QE). The UPLC analytical conditions are as follows: column, Waters ACQUITY UPLC BEH C18 (1.7 μm, 50 × 2.1 mm); column temperature, 55 °C; flow rate, 0.5 mL/min; injection volume, 1 μL; solvent system, water (0.1% formic acid) / acetonitrile (0.1% formic acid); gradient program, 95:5 V/V at 0 min, 90:10 V/V at 5.5 min, 75:25 V/V at 7.5 min, 40:60 V/V at 8 min, 95:5 V/V at 8.5 min, and 95:5 V/V at 13 min.

#### 3.7.3. LC-MS/MS Analysis

The HRMS data were recorded on a Q Exactive hybrid Q–Orbitrap mass spectrometer equipped with a heated ESI source (Thermo Fisher Scientific, Waltham, MA, USA) utilizing the full MS acquisition methods. The ESI source parameters were set as follows: spray voltage, 3 kV; sheath gas pressure, 40 arb; aux gas pressure, 10 arb; sweep gas pressure, 0 arb; capillary temperature, 320 °C; aux gas heater temperature, 350 °C.

### 3.8. Determination of Flavor Metabolites

#### 3.8.1. Sample Preparation and Treatment

The SRW samples were weighed, immediately frozen, and ground to a powder in liquid nitrogen. The powder was stored at −80°C for later analysis. Then, 1 g of the powder was transferred immediately to a 20 mL headspace vial (Agilent, Palo Alto, CA, USA) containing NaCl-saturated solution to inhibit any enzyme reaction. The vials were sealed using crimp-top caps with TFE-silicone headspace septa (Agilent). During SPME analysis, each vial was placed at 60 °C for 5 min. Then, a 120 µm DVB/CWR/PDMS fiber (Agilent) was exposed to the sample’s headspace for 15 min at 100 °C.

#### 3.8.2. GC-MS Analysis

After sampling, the VOCs from the fiber coating were desorbed in the injection port of the GC apparatus (Model 8890; Agilent) at 250 °C for 5 min in the splitless mode. The VOCs were identified and quantified using an Agilent Model 8890 GC and a 7000D mass spectrometer (Agilent) equipped with a 30 m × 0.25 mm × 0.25 μm DB-5MS (5% phenyl-polymethyl siloxane) capillary column. Helium was used as the carrier gas at a linear velocity of 1.2 mL/min. The injector and detector temperatures were kept at 250 and 280 °C. The oven temperature was increased from 40 °C (3.5 min) to 100, 180, and 280 °C at increasing rates of 10, 7, and 25 °C/min, respectively. The final programmed oven temperature was held for 5 min. Mass spectra were recorded in the electron impact ionization mode at 70 eV. The quadrupole mass detector, ion source, and transfer line temperatures were set at 150, 230, and 280 °C, respectively. The selected ion monitoring mode was used to identify and quantify analytes [29,30].

### 3.9. Statistical Analysis

Microsoft Office Excel was used for basic data processing. Moreover, IBM SPSS Statistics 23 data analysis software (IBM Corporation, New York, NY, USA) was used for statistical analysis. The results were expressed as mean ± SD, and the letters after standard deviation indicated significant differences. All experiments were in triplicate. The RSM was used to optimize the rice wine process using Design Expert software 8. An unsupervised principal component analysis (PCA) was performed by the statistics function prcomp within R (www.r-project.org (accessed on 8 August 2022)). The data were unit variance-scaled before the unsupervised PCA. Significantly regulated metabolites between groups were determined by VIP (VIP ≥ 1) and absolute Log2FC (|Log2FC| ≥ 1.0). The VIP values were extracted from the OPLS-DA result, which also contains score plots. Permutation plots were generated using the R package MetaboAnalystR (Github, New York, NY, USA). The data were log transform (log2) and should be centered before OPLS-DA. A permutation test (200 permutations) was performed to avoid overfitting. The identified metabolites were annotated using the KEGG Compound database on.kegg.jp (accessed on 8 August 2022).

## 4. Conclusions

In this study, we optimized the brewing conditions of SRW and compared the effects of thermal and IPL sterilization on the physicochemical properties, free amino acids, and flavor metabolites of the brewed SRW. The RSM study demonstrated that the predicted value of sensory score under the optimized conditions reached 79.6. Moreover, 60 s pulse treatment reduced the bitter amino acids and maintained the sweet and umami amino acids in SRW, resulting in a good effect on the quality of SRW. OPLS-DA analysis showed that 14 kinds of differential flavor metabolites were upregulated under 60 s pulse treatment to enhance the alcohol and esters components in SRW. In general, IPL sterilization can better preserve the volatile flavor components of SRW than pasteurization. This study provides a new method and theoretical basis for the sterilization treatment of SRW. It can also provide reference values for enterprises to produce high-quality SRWs.

## Figures and Tables

**Figure 1 molecules-28-03183-f001:**
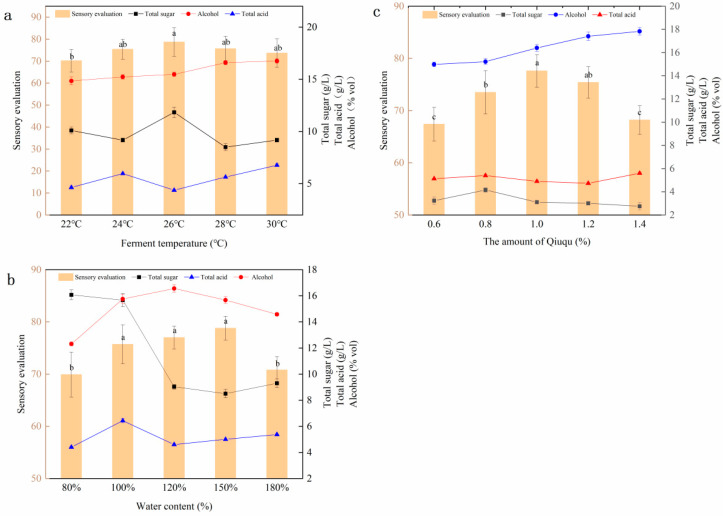
Effect of a single factor on the quality of SRW. (**a**) Effect of fermentation temperature on the quality of SRW, (**b**) effect of water content on the quality of SRW, and (**c**) effect of Qiuqu addition amount on the quality of SRW.

**Figure 2 molecules-28-03183-f002:**
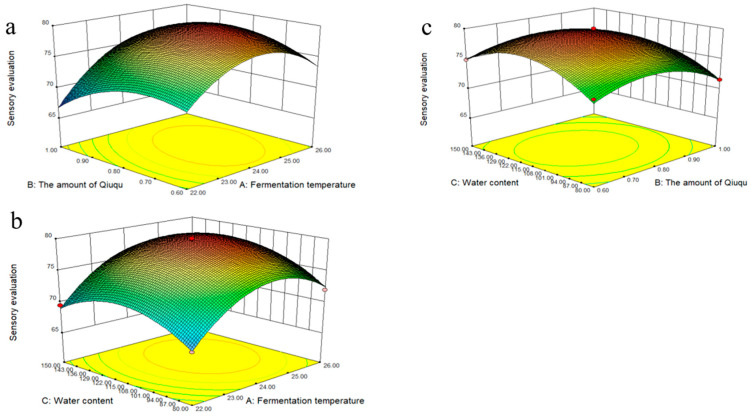
Response surface plot for sensory evaluation as a function of, (**a**) fermentation temperature and the amount of Qiuqu, (**b**) fermentation temperature and water content, and (**c**) water content and the amount of Qiuqu.

**Figure 3 molecules-28-03183-f003:**
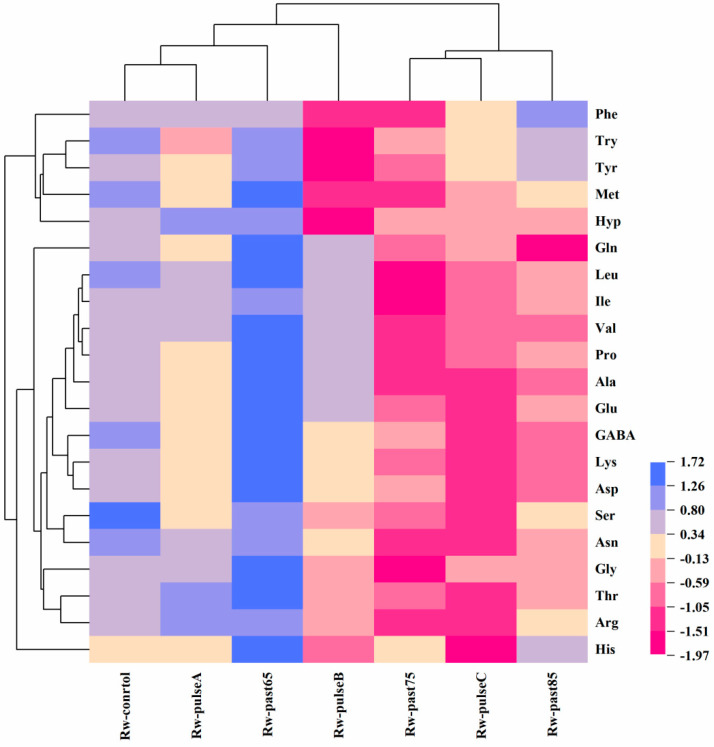
The difference of 21 free amino acids between unsterilized and other sterilized SRW.

**Figure 4 molecules-28-03183-f004:**
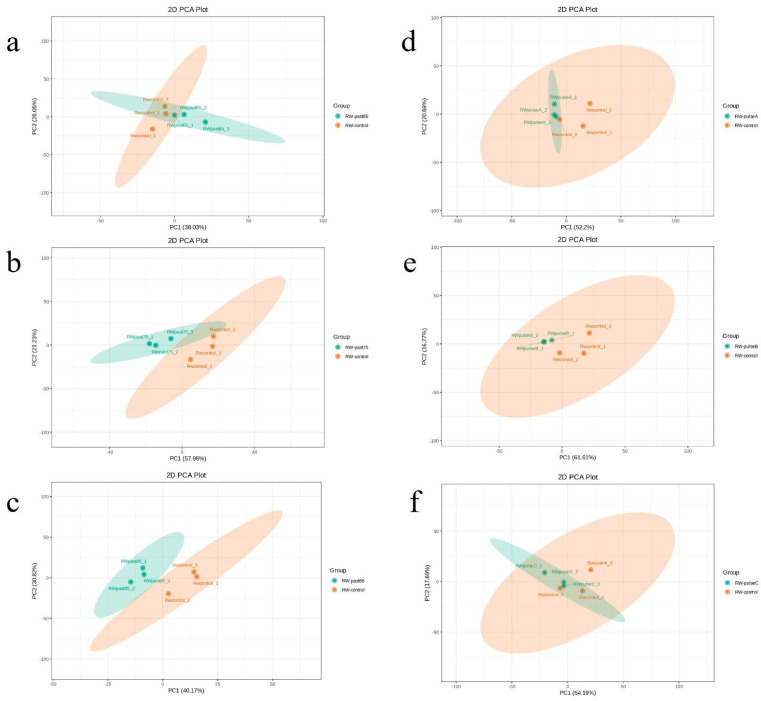
Principal component analysis (PCA) of flavor metabolites in SRW before and after sterilization. (**a**) Rw-control vs. Rw-past65. (**b**) Rw-control vs. Rw-past75. (**c**) Rw-control vs. Rw-past85. (**d**) Rw-control vs. Rw-pulse A. (**e**) Rw-control vs. Rw-pulse B. (**f**) Rw-control vs. Rw-pulse C.

**Figure 5 molecules-28-03183-f005:**
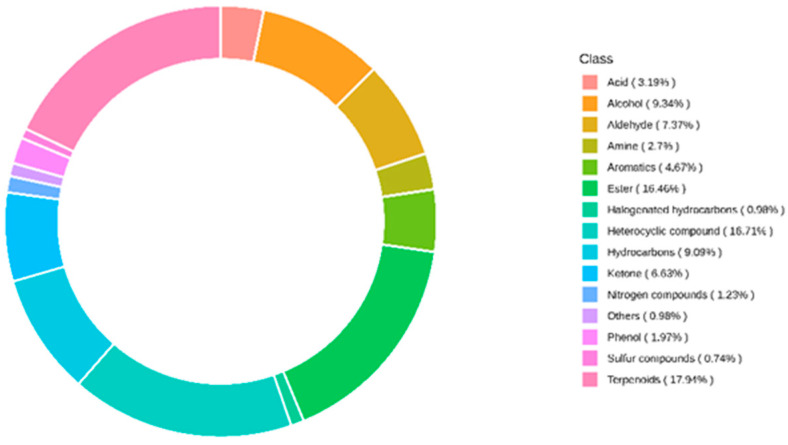
The proportion of the identified flavor metabolites in the composition classification.

**Figure 6 molecules-28-03183-f006:**
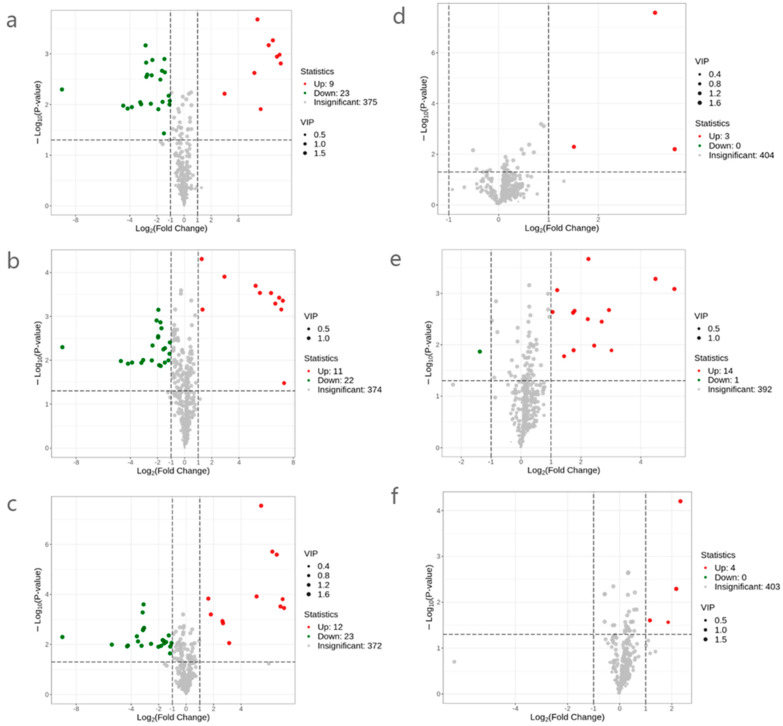
Volcanic map of flavorful metabolites of SRW before and after sterilization. (**a**) Rw-control vs. Rw-past65. (**b**) Rw-control vs. Rw-past75. (**c**) Rw-control vs. Rw-past85. (**d**) Rw-control vs. Rw-pulse A. (**e**) Rw-control vs. Rw-pulse B. (**f**) Rw-control vs. Rw-pulse C.

**Table 1 molecules-28-03183-t001:** Design and result of the response surface experiment.

Run	Factor 1 (X_1_)	Factor 2 (X_2_)	Factor 3 (X_3_)	Response			
	Fermentation Temperature (°C)	The Amount of Qiuqu (%)	Water Content (%)	Sensory Evaluation (Scores)	Total Acid Content (g/L)	Alcohol Content (% (*v*/*v*))	Total Sugar Content (g/L)
1	24	1.00	150.00	72	4.48	12.83	17.76
2	24	0.80	115.00	79	5.23	14.03	15.08
3	26	0.80	80.00	72	6.04	15.61	16.21
4	26	1.00	115.00	75.3	5.54	9.90	14.80
5	22	1.00	115.00	66.9	4.58	13.72	15.36
6	22	0.80	80.00	68	4.92	17.00	16.07
7	24	0.60	80.00	73.8	5.58	17.19	14.66
8	24	0.80	115.00	79.9	5.23	14.40	15.36
9	24	0.80	115.00	79.4	4.97	14.86	14.10
10	26	0.60	115.00	73.6	4.63	14.95	17.06
11	24	1.00	80.00	71.6	5.32	16.11	14.52
12	22	0.80	150.00	69.5	4.26	13.73	15.51
13	22	0.60	115.00	71.3	4.23	8.77	13.67
14	24	0.60	150.00	74.9	4.60	14.61	13.96
15	24	0.80	115.00	79	4.83	14.12	14.80
16	24	0.80	115.00	80	5.02	14.53	14.66
17	26	0.80	150.00	74.7	4.11	13.21	14.94
Model				*p* < 0.0001			
R^2^ (%)				98.99			
R^2^adj (%)				97.70			
R^2^pre (%)				88.58			
Lack of fit				not significant			

**Table 2 molecules-28-03183-t002:** Changes in physicochemical properties of SRW before and after sterilization.

Parameters	Rw-Control	Rw-past65	Rw-past75	Rw-past85	Rw-pulseA	Rw-pulseB	Rw-pulseC
Ethanol (20 °C)/(%, *v*/*v*)	17.53 ± 0.025 ^a^	16.83 ± 0.034 ^b^	16.91 ± 0.054 ^b^	15.33 ± 0.025 ^e^	16.38 ± 0.054 ^c^	16.31 ± 0.033 ^cd^	16.21 ± 0.034 ^d^
Total sugar(g/L)	18.21 ± 0.500 ^c^	20.79 ± 0.995 ^b^	20.44 ± 0.495 ^b^	22.08 ± 0.318 ^a^	17.97 ± 0.518 ^c^	17.82 ± 0.317 ^c^	17.70 ± 0.110 ^c^
Total acid(g/L)	5.29 ± 0.042 ^a^	5.13 ± 0.078 ^bc^	5.23 ± 0.000 ^ab^	5.10 ± 0.042 ^cd^	4.94 ± 0.062 ^e^	4.88 ± 0.047 ^e^	5.00 ± 0.039 ^de^
pH	4.35 ± 0.005 ^c^	4.44 ± 0.009 ^b^	4.36 ± 0.009 ^c^	4.46 ± 0.009 ^b^	4.54 ± 0.009 ^a^	4.52 ± 0.005 ^a^	4.53 ± 0.009 ^a^

Different lowercase letters in the same line represent significant differences (*p* < 0.05).

**Table 3 molecules-28-03183-t003:** Effects of different sterilization conditions on free amino acids in SRW.

Amino Acid (μg/mL)	Rw-Conrtol	Rw-past65	Rw-Past75	Rw-past85	Rw-pulseA	Rw-pulseB	Rw-pulseC
His	16.39 ± 0.51 ^b^	17.45 ± 0.22 ^a^	16.21 ± 0.2 ^b^	16.46 ± 0.3 ^b^	16.32 ± 0.21 ^b^	15.37 ± 0.19 ^c^	14.52 ± 0.39 ^c^
Hyp	2.57 ± 0.1 ^a^	2.65 ± 0.21 ^a^	2.48 ± 0.14 ^a^	2.51 ± 0.14 ^a^	2.63 ± 0.19 ^a^	2.35 ± 0.11 ^a^	2.48 ± 0.19 ^a^
Arg	76.24 ± 0.53 ^c^	79.39 ± 0.16 ^a^	64.5 ± 0.27 ^f^	72.38 ± 0.27 ^d^	78.44 ± 0.07 ^b^	68.51 ± 0.07 ^e^	64.39 ± 0.4 ^f^
Asn	131.56 ± 0.21 ^a^	129.36 ± 0.22 ^b^	108.61 ± 0.25 ^g^	118.53 ± 0.24 ^e^	124.57 ± 0.16 ^c^	122.42 ± 0.22 ^d^	109.31 ± 0.08 ^f^
Gln	309.77 ± 0.82 ^b^	329.47 ± 0.23 ^a^	261.35 ± 0.08 ^f^	243.66 ± 0.26 ^g^	296.71 ± 0.26 ^d^	303.54 ± 0.33 ^c^	286.21 ± 0.01 ^e^
Ser	71.5 ± 0.16 ^a^	69.29 ± 0.19 ^b^	60.22 ± 0.09 ^f^	64.6 ± 0.08 ^d^	65.36 ± 0.19 ^c^	63.36 ± 0.53 ^e^	58.33 ± 0.36 ^g^
Gly	134.07 ± 0.7 ^c^	142.06 ± 0.22 ^a^	121.33 ± 0.28 ^f^	130.45 ± 0.4 ^d^	135.46 ± 0.17 ^b^	130.62 ± 0.29 ^d^	129.49 ± 0.39 ^e^
Asp	56.27 ± 0.25 ^b^	62.76 ± 0.18 ^a^	50.4 ± 0.15 ^e^	47.44 ± 0.29 ^f^	54.45 ± 0.21 ^c^	52.32 ± 0.21 ^d^	45.16 ± 0.23 ^g^
Glu	138.37 ± 0.12 ^b^	152.67 ± 0.16 ^a^	120.34 ± 0.26 ^f^	126.32 ± 0.1 ^e^	133.17 ± 0.12 ^d^	138.39 ± 0.49 ^b^	118.33 ± 0.35 ^g^
Thr	38.58 ± 0.19 ^b^	41.52 ± 0.24 ^a^	34.61 ± 0.23 ^e^	35.55 ± 0.27 ^d^	41.45 ± 0.28 ^a^	36.49 ± 0.41 ^c^	33.24 ± 0.15 ^f^
Ala	348.35 ± 0.24 ^b^	370.23 ± 0.17 ^a^	296.52 ± 0.32 ^g^	308.56 ± 0.1 ^e^	336.27 ± 0.18 ^d^	346.43 ± 0.38 ^c^	299.69 ± 0.26 ^f^
Pro	286.65 ± 0.34 ^b^	301.65 ± 0.34 ^a^	240.17 ± 0.12 ^g^	258.53 ± 0.38 ^e^	277.65 ± 0.13 ^d^	280.33 ± 0.16 ^c^	250.64 ± 0.11 ^f^
Lys	20.74 ± 0.21 ^b^	24.57 ± 0.34 ^a^	15.52 ± 0.16 ^d^	15.67 ± 0.24 ^d^	18.69 ± 0.23 ^c^	18.8 ± 0.15 ^c^	13.39 ± 0.32 ^e^
Met	21.2 ± 0.14 ^b^	21.82 ± 0.15 ^a^	17.5 ± 0.33 ^e^	19.41 ± 0.18 ^c^	19.42 ± 0.13 ^c^	17.34 ± 0.14 ^e^	18.5 ± 0.3 ^d^
Tyr	94.17 ± 0.14 ^b^	97.52 ± 0.31 ^a^	87.44 ± 0.2 ^d^	94.43 ± 0.22 ^b^	91.44 ± 0.41 ^c^	81.2 ± 0.2 ^e^	91.86 ± 0.19 ^c^
Val	64.49 ± 0.24 ^b^	69.51 ± 0.33 ^a^	52.52 ± 0.19 ^g^	56.61 ± 0.22 ^e^	62.73 ± 0.19 ^d^	63.28 ± 0.25 ^c^	55.96 ± 0.16 ^f^
Ile	38.6 ± 0.19 ^b^	40.31 ± 0.11 ^a^	31.44 ± 0.27 ^f^	34.59 ± 0.07 ^d^	37.61 ± 0.3 ^c^	38.34 ± 0.42 ^b^	33.54 ± 0.2 ^e^
Leu	114.69 ± 0.12 ^b^	118.55 ± 0.26 ^a^	95.47 ± 0.14 ^g^	103.31 ± 0.11 ^e^	111.3 ± 0.32 ^d^	111.86 ± 0.09 ^c^	101.5 ± 0.27 ^f^
Phe	82.08 ± 0.15 ^b^	83.66 ± 0.16 ^a^	73.54 ± 1.73 ^c^	84.38 ± 0.23 ^a^	82.48 ± 0.08 ^b^	74.23 ± 0.19 ^c^	81.58 ± 0.11 ^b^
Try	16.24 ± 0.18 ^a^	16.51 ± 0.22 ^a^	14.5 ± 0.16 ^c^	15.82 ± 0.19 ^b^	14.53 ± 0.3 ^c^	11.5 ± 0.4 ^d^	14.61 ± 0.1 ^c^
GABA	82.37 ± 0.46 ^b^	84.62 ± 0.31 ^a^	69.47 ± 0.15 ^d^	66.38 ± 0.21 ^e^	75.3 ± 0.28 ^c^	75.37 ± 0.24 ^c^	62.34 ± 0.19 ^f^
Umani	194.64 ± 0.37 ^b^	215.43 ± 0.3 ^a^	170.74 ± 0.38 ^f^	173.75 ± 0.26 ^e^	187.63 ± 0.3 ^d^	190.71 ± 0.68 ^c^	163.5 ± 0.29 ^g^
Bitter	523.65 ± 1.02 ^b^	547.48 ± 0.95 ^a^	451.15 ± 2.21 ^g^	493.65 ± 1.45 ^d^	513.55 ± 0.31 ^c^	483.09 ± 1.02 ^e^	471.36 ± 1 ^f^
Sweet	864.34 ± 0.7 ^b^	907.7 ± 1.12 ^a^	738.23 ± 0.6 ^g^	784.05 ± 1.02 ^e^	836.79 ± 0.48 ^d^	840.43 ± 0.95 ^c^	759.13 ± 0.74 ^f^
Total	2144.92 ± 3.02 ^b^	2255.57 ± 1.82 ^a^	1834.16 ± 2.57 ^g^	1915.56 ± 3.3 ^e^	2075.98 ± 1.08 ^c^	2052.06 ± 2.86 ^d^	1885.09 ± 1.73 ^f^

Different lowercase letters in the same line represent significant differences (*p* < 0.05).

**Table 4 molecules-28-03183-t004:** Differences of main flavor metabolites in SRW before and after sterilization.

Class		Relative Peak Area						
		Control	Rw-past65	Rw-past75	Rw-past85	Rw-pulseA	Rw-pulseB	Rw-pulseC
Esters	Isobutyric acid, 2-methyl phenyl ester	3064.29 ± 212.338	4021.68 ± 630.041	5252.79 ± 307.514	10,691.62 ± 579.114	3214.03 ± 53.711	4392.76 ± 386.596	3476.62 ± 293.656
	p-Tolyl isobutyrate	12,028.53 ± 205.325	17,559.02 ± 1077.723	29,911.44 ± 1007.936	77,672.37 ± 4326.788	16,982.43 ± 949.484	23,225.27 ± 1166.487	15,345.01 ± 1647.744
	Isopentyl acetate	732,535.5 ± 73,436.897	268,613.25 ± 62,438.254	224,228.23 ± 36,969.183	224,757.95 ± 4961.607	653,504.86 ± 23,454.302	602,725.46 ± 16,781.934	700,803.69 ± 47,184.793
	(E,E)-2,6,10-Dodecatrienoic acid, 3,7,11-trimethyl-, methyl ester	13,613.64 ± 3988.728	4177.8 ± 115.78	6463.12 ± 2022.43	4647.21 ± 86.049	14,803.81 ± 588.338	18,140.12 ± 1476.511	14,469.59 ± 545.758
	(E)-9-Tetradecen-1-ol, acetate	17,606.21 ± 5413.779	5924.86 ± 969.367	8907.79 ± 2279.311	6641.17 ± 601.302	19391.19 ± 421.39	23,789.08 ± 1184.703	19,636.87 ± 723.008
	(E)-3, 7-dimethyl octyl-2, 6-dienyl 2-methyl butyrate	51,110.66 ± 4382.248	7797.53 ± 838.508	13,219.06 ± 1108.526	5950.76 ± 810.254	52,802.61 ± 5542.946	85,948.53 ± 18,498.518	50,599.68 ± 2327.055
	Ethyl octanoate	1,949,717.16 ± 336,693.747	87,137.31 ± 19,088.494	75,100.86 ± 16,436.007	45,943.67 ± 17,635.828	1,896,875.12 ± 40,543.238	1,885,887.78 ± 39,710.295	2,090,866.82 ± 36,486.562
	Butyric acid hexyl ester	156,814.59 ± 28,451.425	8,762.96 ± 1,075.5	8,678.03 ± 967.986	7,991.14 ± 826.559	151,275.06 ± 4,584.326	150,082.6 ± 4103.551	165,890.7 ± 3696.6
	Pentanoic acid, 4-methyl-methyl ester	29,881.13 ± 3168.636	13,648.63 ± 1157.526	13,129.85 ± 4741.83	12,470.63 ± 1413.552	25,590.93 ± 2968.793	28,316.2 ± 7503.432	25,375.05 ± 4840.742
	Resorcinol monoacetate	8748.93 ± 964.15	2835.26 ± 545.963	4046.45 ± 372.45	3934.37 ± 119.828	8296.5 ± 239.1	9026.14 ± 516.486	8241.96 ± 957.017
	Octyl butyrate	257,667.55 ± 39,485.494	27,468.92 ± 4380.501	31,012.32 ± 1347.491	22,783.38 ± 7640.48	256,052.98 ± 7341.198	299,970.59 ± 6593.387	265,885.75 ± 16,229.577
	Total	3,232,788.19	447,947.22	419,949.94	50,942.57	3,098,789.52	3,131,504.53	3,360,591.74
Alcohols	2,6-Dimethyl-1-nonen-3-yn-5-ol	4399.45 ± 776.729	4959.92 ± 567.986	6977 ± 631.921	13,534.5 ± 807.302	5389.07 ± 1160.899	5586.05 ± 519.566	5091.27 ± 211.157
	2-Ethyl-1-dodecanol	59,375.02 ± 5554.812	86,45.81 ± 1189.908	15,278.39 ± 1610.646	6733.12 ± 1159.443	58,705.96 ± 4355.568	98,319.92 ± 21,247.89	57,452.67 ± 3117.81
	(1α, 2α, 3α)-2-methyl-3-(1-methylethenyl)-Cyclohexanol	132,292.58 ± 15,869.088	44,087.29 ± 2911.496	48,112.85 ± 1153.987	42,473.48 ± 4229.971	142,957.67 ± 2669.479	138,343.46 ± 5552.434	146,039.16 ± 5189.607
	(Z)-3-nonyl-1-alcohol	855.88 ± 877.617	113,375.52 ± 6817.513	118,909.62 ± 6030.446	112,044.97 ± 3506.555	2119.61 ± 566.665	6952.1 ± 1740.731	3102.97 ± 534.152
	1,3-Dioxolane-2,2-diethanol	4628.89 ± 570.668	ND	ND	ND	4548.44 ± 233.204	5949.89 ± 441.628	4315.01 ± 151.088
	Total	201,551.82	171,068.54	189,277.86	174,786.07	213,720.75	255,151.42	216,001.08
Ketones	1-Hepten-3-one	349,577.97 ± 39,562.94	131,182.08 ± 26,877.715	118,109.34 ± 12,500.191	115,758.5 ± 3226.809	315,648.35 ± 28,105.849	299,848.98 ± 16,709.773	344,682.67 ± 18,267.568
	3,4-Hexanedione, 2,2,5-trimethyl-	6867.29 ± 2424.017	4476.04 ± 552.765	4126.52 ± 818.708	7201.94 ± 1724.076	19,556.12 ± 2959.43	32,710.9 ± 1738.292	15,332.5 ± 301.281
	4-phenyl-2-butanone	3161.09 ± 306.501	3675.17 ± 476.072	3130.46 ± 214.618	4501.33 ± 671.979	5915.59 ± 381.984	10,682.54 ± 1619.663	4731.09 ± 121.618
	2,2,6-trimethyl-cyclohexanone	76,828.05 ± 6918.019	40,305 ± 8873.589	36,174.55 ± 2929.82	40,554.89 ± 3659.857	76,779.21 ± 2724.664	82,392.47 ± 6402.953	73,773.66 ± 8257.353
	Total	436,434.40	179,638.29	161,540.87	168,016.66	417,899.27	425,634.89	438,519.92
Phenols	3-methyl-phenol	1664.79 ± 293.884	2639.11 ± 1060.4	2149.54 ± 572.866	2390.39 ± 661.082	19,278.33 ± 2542.569	58,085.63 ± 2950.302	8369.55 ± 144.182
	p-Cresol	3059.76 ± 183.622	3952.55 ± 1599.361	3910.41 ± 290.461	3312.75 ± 400.071	26,949.13 ± 224.902	68,871.51 ± 2677.255	13,764.45 ± 1407.483
	Total	4724.55	6591.66	6059.95	5703.14	46,227.46	126,957.14	22,134.00
Acids	alpha-cyclopentyl-Benzeneacetic Acid	730.98 ± 659.929	36,857.73 ± 7172.521	117,134 ± 37,731.265	47,504.99 ± 20,360.309	960.12 ± 885.407	579.68 ± 496.721	ND
	Total	730.98	36,857.73	117,134.00	47,504.99	960.12	579.68	ND
Aldehydes	(2E,4Z)-2,4-Decadienal	11,583.35 ± 2596.763	5398.78 ± 434.088	8227.6 ± 345.334	18,593.62 ± 192.731	18,232.51 ± 3852.31	15,761.81 ± 1774.755	14,633.17 ± 906.357
	7-methyl-3-methylene-6-octenal	4859.97 ± 4396.688	562,199.41 ± 35,086.469	606,166.83 ± 24,301.727	568,750.45 ± 21,586.876	6901.12 ± 1117.369	31,490.25 ± 5610.622	10,932.13 ± 1283.039
	Benzaldehyde	233,259.01 ± 39,542.699	258,615.46 ± 29,804.367	234,231.83 ± 3959.265	298,238.85 ± 8346.082	380,206.41 ± 32,780.122	538,673.48 ± 24,656.753	343,914.71 ± 49,818.112
	Total	249,702.33	826,213.65	848,626.26	885,582.92	405,340.04	585,925.54	369,480.01

## Data Availability

Not applicable.

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
