# Peer review of "Optimization of the Brewing Conditions of Shanlan Rice Wine and Sterilization by Thermal and Intense Pulse Light"

_molecules, 2023, doi:10.3390/molecules28073183_

Round 1

Reviewer 1 Report

Manuscript Number: molecules-2282957

Title: Optimization of the brewing conditions of Shanlan rice wine and sterilization by thermal and intense pulse light

The authors correctly planned the experiment to optimize the fermentation process of Shanlan rice wine (SRW). In addition, they checked how thermal pasteurization and intense pulsed light (IPL) processes affect the physicochemical properties of SRW, free amino acids in SRW and flavor metabolites in SRW. The data analysis methodology used is typically used in this type of research and is correctly cited. The manuscript is written in plain, understandable language.

In my opinion, the weak side of the manuscript is the lack of research on the effects of the sterilization used. I don't know why sterilization was used. Therefore, the following questions remain unanswered: Did the given sterilization extend the shelf life of the product? Has it improved its quality? Can the product be stored at ambient temperature or does it need to be cooling? The SRW sampes was filtered and centrifuged, sow was sterilization a necessary process at all?

Minor revisions and comments:

1.     p16, line 435 it is ‘SRA’ should be replaced with ‘SRW’, if not please explain what the abbreviation SRA stands for

2.     in abstract, line 11, please expand shortcuts LC-MS/MS and GC-MS

3.     in abstract, line 14, the word Qiuqu appears for the first time - it should be added that it is the name of active form of the traditional yeast used to produce SRW

4.     p16, subsection “4.5. Sterilization of SRW” IPL sterilization was performed with a xenon lamp. Please provide the model and manufacturer of the lamp, it is also necessary to provide the power of the light used.

Author Response

1. Did the given sterilization extend the shelf life of the product? Has it improved its quality? Can the product be stored at ambient temperature or does it need to be cooling? The SRW sampes was filtered and centrifuged, sow was sterilization a necessary process at all?

Answer:  Sterilization can prolong the shelf life of Shanlan rice wine. The research on the specific shelf life of Shanlan rice wine is in progress; The sterilization treatment of Shanlan rice wine can reduce the harmful microorganisms in the wine. Meanwhile, the treatment with intense pulse light 60s can improve the flavor of Shanlan rice wine, reduce the contents of bitter amino acids and maintain the contents of sweet and umami amino acids, so as to improve the quality of the product and the safety of the microorganism; The SRW can be stored at ambient temperature after sterilization. Centrifugation and filtration only can remove impurities and precipitate in Shanlan rice wine, but sterilization can reduce harmful microorganisms in wine and improve the safety of products.

2. p16, line 435 it is ‘SRA’ should be replaced with ‘SRW’, if not please explain what the abbreviation SRA stands for

Answer: Changes have been made in the manuscript.

3.in abstract, line 11, please expand shortcuts LC-MS/MS and GC-MS

Answer: It has been supplemented in the manuscript.

4. in abstract, line 14, the word Qiuqu appears for the first time - it should be added that it is the name of active form of the traditional yeast used to produce SRW

Answer: It has been supplemented in the manuscript.

5. p16, subsection “4.5. Sterilization of SRW” IPL sterilization was performed with a xenon lamp. Please provide the model and manufacturer of the lamp, it is also necessary to provide the power of the light used.

Answer: The model number and manufacturer of the pulse equipment have been added to the manuscript. The xenon lamp manufacturer has not provided the model number. The intensity of xenon lamp operation is defined in terms of output light intensity, which has been supplemented.

See the attachment for the manuscript.

Reviewer 2 Report

The manuscript mainly described the optimization of the brewing parameters and sterilization methods of Shanlan rice wine. This topic is interesting and meaningful. However, something puzzling problems are existed in the Manuscript and should be addressed.

1. Single-factor experiment results revealed that optimum fermentation parameters were temperature of 26 °C, Qiuqu amount of 1%, and water content of 150% (Figure 1) based on sensory evaluation. For the RSM experiments, variables were coded at three levels (-1, 0, 1), and the center points coded as zero were temperature of 24 °C, Qiuqu amount of 0.8%, and water content of 115% (Table 1). Why?

2. Sensory Evaluation Method should be further expanded especially for the sensory score. Did authors use any standard compound/method for scoring?

Author Response

1. Single-factor experiment results revealed that optimum fermentation parameters were temperature of 26 °C, Qiuqu amount of 1%, and water content of 150% (Figure 1) based on sensory evaluation. For the RSM experiments, variables were coded at three levels (-1, 0, 1), and the center points coded as zero were temperature of 24 °C, Qiuqu amount of 0.8%, and water content of 115% (Table 1). Why?

Answer: First, fermentation temperature. Considering that the growth temperature of yeast is generally controlled at 25℃, and that there is little difference in sensory scores between 22℃ and 28℃ in the single factor experiment, 22℃, 24℃ and 26℃ are selected for the following response surface experiment after comprehensive consideration. Secondly, with regard to the amount of Qiuqu added, the brewing conditions of Shanlan rice wine finally obtained in this study may be applied to actual factory production. Considering the production cost, 0.6%, 0.8% and 1.0% were finally selected for the following response surface experiment. Finally, regarding water content, in the single-factor experiment, although the sensory score of 150% water content is higher, its color is poor, while the color of 100% water content and 120% water content are better and there is no significant difference between the sensory scores, so 100% water content and 120% water content are considered as the optimal value.

2. Sensory Evaluation Method should be further expanded especially for the sensory score. Did authors use any standard compound/method for scoring?

Answer: The sensory evaluation was conducted with the official methods of GB/T 13662-2018 in China with some modifications. Specific instructions have been added accordingly in the article.